# Neurocognitive Psychiatric and Neuropsychological Alterations in Parkinson’s Disease: A Basic and Clinical Approach

**DOI:** 10.3390/brainsci13030508

**Published:** 2023-03-18

**Authors:** Héctor Alberto González-Usigli, Genaro Gabriel Ortiz, Claudia Charles-Niño, Mario Alberto Mireles-Ramírez, Fermín Paul Pacheco-Moisés, Blanca Miriam de Guadalupe Torres-Mendoza, José de Jesús Hernández-Cruz, Daniela Lucero del Carmen Delgado-Lara, Luis Javier Ramírez-Jirano

**Affiliations:** 1Department of Neurology, Clinic of Movements Disorders, High Specialty Medical Unit, Western National Medical Center of the Mexican Institute of Social Security, Guadalajara 44340, Mexico; 2Department of Philosophical and Methodological Disciplines, University Center of Health Sciences, University of Guadalajara, Guadalajara 44340, Mexico; 3Department of Microbiology, University Center of Health Sciences, University of Guadalajara, Guadalajara 44340, Mexico; 4Department of Chemistry, University Center of Exact Sciences and Engineering, University of Guadalajara, Guadalajara 44430, Mexico; 5Division of Neurosciences, Western Biomedical Research Center, Mexican Institute of Social Security, Guadalajara 44340, Mexico

**Keywords:** Parkinson’s disease, pathophysiology, genetics, psychiatric illness, cognitive dysfunction, neuropsychological tests

## Abstract

The main histopathological hallmarks of Parkinson’s disease (PD) are the degeneration of the dopaminergic neurons of the substantia nigra pars compacta and the loss of neuromelanin as a consequence of decreased dopamine synthesis. The destruction of the striatal dopaminergic pathway and blocking of striatal dopamine receptors cause motor deficits in humans and experimental animal models induced by some environmental agents. In addition, neuropsychiatric symptoms such as mood and anxiety disorders, hallucinations, psychosis, cognitive impairment, and dementia are common in PD. These alterations may precede the appearance of motor symptoms and are correlated with neurochemical and structural changes in the brain. This paper reviews the most crucial pathophysiology of neuropsychiatric alterations in PD. It is worth noting that PD patients have global task learning deficits, and cognitive functions are compromised in a way is associated with hypoactivation within the striatum, anterior cingulate cortex, and inferior frontal sulcus regions. An appropriate and extensive neuropsychological screening battery in PD must accurately assess at least five cognitive domains with some tests for each cognitive domain. This neuropsychological screening should consider the pathophysiological and clinical heterogeneity of cognitive dysfunction in PD.

## 1. Introduction

In 1817, the English physician James Parkinson published the first clinical essay on “paralysis agitans”, a term by which he defined the disease, and he reported six cases characterized by involuntary tremor movement with decreased muscular strength in passive and active mobility and with an increased forward inclination of the trunk [1]. PD is the second most common neurodegenerative disease after Alzheimer’s [2]. Its neurological basis remained unknown for more than a hundred years, while histopathological examinations of brains with PD showed an significant loss of nerve cells of the substantia nigra. The term substantia nigra derives from the fact that its cells appear dark due to the neural pigment in them called neuromelanin. These pigmented cells are lost in PD, and an even more critical aspect is that this set of cells uses dopamine as their primary neurotransmitter. This cell degeneration is accompanied by a severe reduction (more than 80%) in the dopamine content in the striatum (caudate nucleus and putamen), the point of greatest termination of the axons from the substantia nigra [3].

Progressive damage to the ascending dopaminergic system, in particular to the nigrostriatal pathway, which is mainly responsible for the motor disorders of PD, is based on the following:Motor parkinsonian symptoms begin to appear when striatal dopamine levels are reduced between 70% and 80% of normal levels, which approximately corresponds to a loss of 50% of the total synapses of the substantia nigra towards the striatum, which is the central nucleus of input of motor information from the cortex towards the basal nuclei, fed back by nigrostriatal pathways to modulate movement through motor anagrams (chunks of motion information) stored in the basal ganglia.This threshold is directly related to the appearance of symptoms: there is a correlation between the extent of damage to the dopaminergic system and the severity of the symptoms because neuronal destruction gradually produces a progressive deficit of dopamine in the striatum, which induces a significant loss of voluntary or involuntary spontaneous movement.Destruction of the striatal dopaminergic pathway and blocking of striatal dopamine receptors cause motor deficits similar to those observed in PD, both in humans and experimental animals induced with 1-methyl-4-phenyl-1,2,3,6-tetrahydropyridine (MPTP), 6-hydroxydopamine (6-OH-DA), reserpine, or methamphetamine.Drug therapies that cause an increase in dopamine availability or stimulate dopamine receptors at the level of the striatum reduce the symptoms associated with this deficit, mainly motor ones [4].

PD affects the dopaminergic system and the serotonergic, glutamatergic, and GABAergic. This results in motor disorders, but also various non-motor functional diseases. The main typical symptoms and definition of PD are a deceleration in movement (bradykinesia) as well as increased muscle tone (rigidity), and tremors at rest [5]. The axial bradykinesia and rigidity are associated with the typical posture, gait, and balance issues that affect PD patients over time. In recent years, more attention has been paid to non-motor symptoms since they are equally crucial in disability and reduced quality of life. These include psychiatric disorders such as depression, anxiety, apathy, obsessive compulsive problems, sleep disorders, cognitive limitations (dysexecutive syndrome, dementia), and functional autonomic disorders: gastrointestinal, urological, sexual, orthostatic, and other vegetative symptoms [6].

PD cannot be reduced to its motor manifestations. In recent years, interest in the neurocognitive, psychiatric, and neuropsychological disorders of PD has developed considerably. PD represents a preponderant area of neuropsychiatry for symptomatic manifestations and neurophysiopathology. Some people experience depressive syndromes before the onset of motor symptoms; others develop psychiatric symptoms or syndromes simultaneously or after the beginning of motor symptoms. Others may develop iatrogenic behavioral disorders, such as dopaminergic dysregulation syndrome, punding associated with levodopa, and impulse control disorders underlying dopamine agonist intake [7,8]. In this case, the clinician must know about pathological gambling, buying, collecting, and excessive eating, and their differential diagnoses.

The most frequent non-motor symptoms in PD are depression, anxiety, apathy, sleep disturbances, psychotic features, behavioral changes, and cognitive deficits [9]. For patients and relatives, these disorders are often more distressing and disabling than the motor aspects. The presence of depressive symptoms seems to be related to an earlier drop in dopaminergic synaptic power or associated iatrogenically with an abrupt discontinuation of dopaminergic therapy, as well as more significant functional disability, more rapid physical and cognitive decline, higher mortality, worsening quality of life, and greater suffering for caregivers. Depressive disorders are often underdiagnosed and, even if identified, are often not adequately treated [10]. The clinical overlap of some depressive and anxious dimensions with symptoms typical of parkinsonian pathology, with possible cognitive alterations and somatic symptoms, further complicates diagnosing neuropsychiatric symptoms. Symptoms of depression, such as insomnia, weight loss, psychomotor retardation, and energy loss, are often attributed to a neurological disorder rather than a concomitant mood disorder. The frequency of depressive disorders in PD is approximately 40–50% [11].

However, the prevalence of depressive symptoms differs according to the clinical population studied, the depressive subtypes evaluated, and the heterogeneity in the syndromic presentation. Several prevalence studies have shown that less than half of patients reporting depressive symptoms have a definite diagnosis of major depression. The prevalence of dysthymia and minor depression is 22.5% and 36.6%, respectively [7]. For this reason, the term subsyndromic depression is introduced as the presence of depressive symptoms that do not fully meet the standardized diagnostic criteria for the different depressive disorders; for example, patients who display depressive symptoms only during “off” states may fit this definition [12]. Several studies on the severity of depressive symptoms have reported that many patients with PD present moderate to severe depressive symptoms. In addition, these patients develop other comorbid neuropsychiatric symptoms, such as anxiety disorders, cognitive deficits, and psychosis [13]. The cardinal symptoms of depression in PD, unlike primary depressions, mainly include fatigue, agitation, psychomotor retardation, apathy, motor retardation, hypomimia, and difficulty concentrating. Depressive symptoms can appear at any stage in the course of the disease, but, on average, they precede the onset of PD by between 4 and 6 years. In addition, the presence of depressive symptoms and any pharmacological treatment influence the evolution of motor symptoms over time. The main risk factors contributing to depression in PD include motor disability, female gender, and previous psychiatric history.

After using deep brain stimulation (DBS) to treat motor symptoms, mood disturbances provide another window to study depression in PD. The risk of suicide, aggression, depression, and mania are among the most frequent psychiatric and behavioral complications after DBS treatment [14]. Indeed, it is hypothesized that subthalamic stimulation inhibits serotonergic transmission through the interconnections between the substantia nigra pars reticulata, medial prefrontal cortex, and ventral pallidum [15]. The presence of anxiety disorders in patients with PD is well documented. Although anxiety is often associated with depression, it can manifest independently and significantly affect patients’ quality of life; the prevalence rates shown range from 25% to 52% of patients [16].

Psychotic symptoms in PD are characterized mainly by hallucinations (primarily visual), delusions, and other sensory disturbances and occur in 20–40% of cases, negatively affecting patients’ quality of life. Psychotic symptoms are also often attributable to the use of antiparkinsonian drugs; however, it is increasingly recognized that the underlying pathological process of PD plays an essential role in its pathogenesis and expression [17,18].

Cognitive, psychiatric, and neuropsychological disorders are essentially part of the alteration of the quality of life of patients and their caregivers. Depressive and anxiety syndromes, hallucinations and other “psychotic” symptoms, apathy, and impulse control disorders raise difficult pathophysiological questions and discussions around the factors related to premorbid vulnerability, disease, and its pharmacological or surgical treatments.

The present work aimed to contribute to the knowledge of non-motor symptoms of PD, highlighting the pathophysiological factors involved in the origin of PD, the superior functions affected, the neuropsychological alterations, and the main tests to evaluate them.

## 2. Pathogenic Factors

PD should be managed more like a syndrome, since pathological mutations are well-implicated with the disease in an autosomal dominant or a recessive way. Nevertheless, other mutations can generate lower susceptibility to PD to some extent, most likely depending on exposure to toxic or biological environmental agents that have not yet been well identified. The accumulation of different proteins in the central nervous system can express this. In this sense, we understand PD as all those diseases or conditions that have a common final effect: a significant reduction in the number of neurons in the substantia nigra and a progressive deficit of dopamine (which we know as presynaptic parkinsonism, where a main feature is a sustained and surprising responsiveness to levodopa treatment over time, regardless of its cause or the proteins that accumulate associated with its presentation). Future research and the demonstration of its relationship with specific genetics or environmental agents will truly clarify the diseases that make up this parkinsonian syndrome (Parkinson’s syndrome). A current theory suggests that dopaminergic neurons are damaged and eventually die due to chronic oxidative stress. The uncontrolled generation of reactive oxygen species (ROS), such as anion superoxide, peroxynitrite, hydrogen peroxide, etc., causes severe damage to nucleic acids, proteins, carbohydrates, and lipids [19]. The neurons of the substantia nigra are susceptible to oxidative stress in PD [20]. Environmental toxins such as paraquat, MPTP, and rotenone can destroy dopaminergic neurons currently used in animal models of PD [21] (Figure 1).

Histopathologically, PD is primarily characterized by abnormal accumulation and deposition of aggregated alpha-synuclein in dopaminergic neurons of the central nervous system (CNS) in the form of intracellular inclusions, the “Lewy bodies”. This leads to a loss of dopaminergic neurons at the level of the substantia nigra, and, incidentally, to a lack of dopamine in the putamen and the caudate nucleus (striatum).

Given the variable clinical picture, a combination of the following pathogenic factors should be considered:Genetic factors: The onset of disease before age 40 indicates a parkinsonian syndrome of genetic origin, the most typical representative of which is the autosomal recessive mutation of Parkin (PARK2). Monogenic families represent less than 5%. More than 15 genetic mutations have been reported with autosomal dominant or recessive inheritance. However, family history comes into play, even in monogenic cases, as the risk of PD is 6.7 times higher for siblings and 3.2 times for children of patients with PD [22].Metabolic factors: Metabolic alterations leading to chronic oxidative stress, mitochondrial dysfunction, and abnormal endo- and exotoxin elimination mechanisms are postulated.Environmental factors: Exposure to certain environmental toxins such as heavy metals (mainly manganese but also copper, lead, or iron), welding (or work with the use of welding material), insecticides, herbicides, and others represent epidemiologically proven risk factors.

The early detection of abnormal alpha-synuclein deposits also in gastrointestinal extracerebral neurons (plexuses and ganglia) in PD patients gives new support to the exotoxic hypothesis. This, and the observation of the cauda–cranial dissemination of cognitive disorder with Lewy bodies in neurons of the CNS, beginning in the olfactory bulb and the neurons of the caudal part of the nucleus of the vagus nerve, constitute elements in favor of the entry of a causative agent through the nasal membrane or intestinal mucosa (gastrointestinal hypothesis) [23]. Additionally, PD could be another disease caused by an abnormally folded protein that spreads similarly to prions [24]. This hypothesis has recently caused a strong surge in PD research; while these data have not been fully demonstrated, recently, the gastrointestinal microbiome of PD patients made establishing a relationship with PD possible [25] (Figure 2).

The incidence of this disorder in western countries is around 200 cases per 100,000 people (depending on the country), most of them over 60 years of age. The main symptoms of PD involve movement and include tremors, stiffness, bradykinesia, and postural disturbances. PD is not just a motor disorder, since many patients show deficits in cognitive function up to dementia with severe impairment of memory, abstract thinking, and language. The presence of affective and behavioral disorders varies from 12 to 90% of cases, and frames PD as a neuropsychiatric disorder.

## 3. Pathophysiology of Neuropsychiatric Alterations in Parkinson’s Disease

The diminution of dopaminergic neurons in the nigrostriatal pathway is reflected in the functioning of four frontostriatal circuits connected in motor, cognitive, affective, and motivational aspects [26]. Two of these circuits are of particular importance in the study of cognitive aspects in PD: (1) the dorsolateral circuit, which comprises the dorsolateral prefrontal cortex, the striatum at the dorsolateral caudate level, the globus pallidus at the dorsomedial level, and the thalamus; and (2) the orbital circuit, which includes the orbitofrontal cortex, the striatum at the level of the ventromedial caudate, the globus pallidus at the dorsomedial level, and the thalamus [27]. Within each of these frontostriatal circuits, two pathways project from the striatum for different exit areas of the basal ganglia and the thalamus towards the prefrontal cortex:The direct pathway (dorsal striatum/internal segment of the globus pallidus/pars reticulata of the substantia nigra/thalamus/premotor cortex/orbitofrontal cortex) exerts a facilitating action of movement.The activity of the indirect pathway (dorsal striatum/external segment of the globus pallidus/internal segment of the globus pallidus/pars reticulata of the substantia nigra/thalamus/premotor cortex/orbitofrontal cortex) has a modulatory inhibition of the action of rival movements to the specific tasks the basal ganglia have chosen to do [28]. Therefore, the deficit of cognitive functions based on the prefrontal cortex, defined globally as an executive deficit (attention, executive roles, working memory), which characterizes many patients with PD from the early stages of the disease, does not derive so much from a direct pathology of the prefrontal cortex as from the reduction in dopaminergic stimulation at the striatal level, which prevents the normal functioning of the frontostriatal circuits. Recent neuroanatomical studies suggest that the evolutionary profile of executive deficit in PD follows the spatiotemporal progression of dopamine reduction at the striatal level concerning the different frontostriatal connections [29]. In the early stages of the disease, dopamine depletion occurs mainly in the dorsolateral portion of the caudate nucleus, an area connected to the dorsolateral prefrontal cortex [30], and, thus, affects the dorsolateral frontostriatal circuit [26]. Therefore, in the initial stages of the disease, the executive functions based on this dorsolateral circuit will be deficient, while the executive functions based on the orbital circuit, not yet affected by the dopaminergic reduction, will be mostly intact. The different levels of dopaminergic reduction between the dorsolateral and the orbital circuits and the distinction between direct and indirect pathways within each frontostriatal circuit are of particular importance.

According to the evolutionary theory of Paul MacLean (1973) of the so-called “tripartite brain”, the structures of the brainstem (hypothalamus, thalamus, and basal nuclei), which characterize the brain of ancestral reptiles, belong to the section of the reptilian brain [31]. This part of the brain is responsible for the perceptual function, the sensory–motor activity that allows the discrimination and generalization of environmental objects to satisfy reproductive and metabolic needs. This is expressed thanks to the control this system exerts on the vegetative functions and the striated muscles that give rise to primordial emotions or underlying emotions. One of the critical functions of the basal ganglia is to filter the commands of voluntary movements that originate in the motor cortex. The default state of the basal nuclei consists of a signal of no movement. Nevertheless, when there is influence through the frontal cortex on the need to perform a task or several tasks, the nuclei of the base are the filters through which their circuits lead to the automatic decision-making of the motor anagrams necessary to perform only one task, dual tasks, or multiple tasks, which is a state different from its basal state of braking all types of voluntary and involuntary movement, except breathing, which has its autonomous control in the nuclei of the brain stem (bulb and bridge) [32]. In a certain sense, the activity of dopaminergic projections in the striatum helps to keep the “door open” to be able to execute a movement. In contrast, the loss of this activity alters the “opening of the door”, making it difficult for the individual to initiate voluntary movements [33] (Figure 3).

Dopamine and norepinephrine are the main catecholaminergic transmitters in the brain. These substances are synthesized through various steps and are then stored in synaptic vesicles for later release. The release process is controlled by inhibitory autoreceptors located in the cell body, dendrites, and catecholaminergic neurons’ terminals. Catecholamines are inactivated by reuptake from the synaptic wall and by enzymatic degradation [34].

The cells of the substantia nigra synthesize the neurotransmitter dopamine, whose stages, which occur at the dopaminergic synapse, are:(a)synthesis of dopamine from tyrosine;(b)accumulation of dopamine by the reserve granules;(c)dopamine release;(d)interaction with its receptor;(e)synaptic reactivation (reuptake) for the subsequent metabolization inactivation (Figure 4).

Certain drugs can modify the catecholaminergic function by acting on the synthesis, release, reuptake, and metabolism processes. These compounds are used clinically to treat various diseases [35].

In the early 1960s, the location in the brain of nerve cells and fibers containing dopamine and norepinephrine were mapped, and a classification system of catecholaminergic cells was proposed using the letter A plus a numbering from 1 to 16 [36]. Of particular importance are the group of A9 cells associated with the defined structure of the substantia nigra and the A10 group, which is found in a very close area called the ventral tegmental area (VTA), which, although it is predominantly dopaminergic, does not contain neuromelanin. It also contains GABAergic and glutamatergic neurons in the central portion between both nigro substances.

The ascending dopaminergic system can be divided into three pathways:The nigrostriatal pathway, which originates in the substantia nigra (group of cells A9) and innervates the caudate putamen (striatum).The mesolimbic pathway, which originates in the VTA (A10 cell group) and innervates various limbic system structures such as the nucleus accumbens, the hippocampus, the lateral septum, and the amygdala.The mesocortical pathway, which originates in the VTA and innervates the cerebral cortex.

To examine the role of catecholamines in behavior, researchers sometimes damage these systems in animals and then evaluate the resulting functional changes.

Neurotransmitters influence many aspects of arousal, attention, the sleep–wake cycle, memory, and affective behaviors by modulating the activity of neurons in the cerebral cortex. The noradrenergic system modulates cortical functions, especially in stressful situations. The role of the basal ganglia (due to its multiple connections with the cortex, the limbic system, the midbrain, and the thalamus) is to integrate motor behaviors with cognitive and emotional components. Therefore, neurotransmitters play a fundamental role in motor and postural coordination, in executive functions (production, maintenance, and modification of motor programs, but also emotional and cognitive ones), in the control of spontaneous emotional expressions, in social behavior, in procedural memory, in language, in spatial attention, and in working memory [37].

Mild cognitive deficits are already present in de novo patients, that is, in the appearance of the clinical motor picture that leads to diagnosing PD, especially in memory and frontal functions [38,39]. Patients with PD also present psychomotor slowing down, with slower information processing speeds and longer reaction times [40], independent of motor disorders [41]. Lewy body dementia and PD-associated dementia share the same neuropathology in the presence of Lewy bodies and cytoplasmic neuronal inclusions. Lewy body dementia [42], characterized by parkinsonism, visual hallucinations, and cognitive fluctuations, differs from dementia associated with PD in the amount of cholinergic deficit (more significant in dementia associated with PD), in the accumulation of beta-amyloid (a common pathophysiological hallmark of Alzheimer’s disease), and in the rapid progression towards cognitive deficits in the first three years of the disease (the appearance of dementia within the first year is only used for research purposes by the high specificity that is clinically conferred to it) [43].

The pathophysiological theory of Hughlings Jackson exhibits how a lesion in a system with hierarchical dynamic structuring and organization determines two types of effects: (a) negative, which causes hypoactivity of functions by direct consequence or hyperactivity with consequent functional alteration; and (b) positive, following the indirect consequences of the injury in the areas connected to functional levels other than the injured one. Therefore, there would be a process of local dissolution or disintegration with alterations in the lower levels of integration and disturbances of the integrated or instrumental functions and a process of uniform or global dissolution with alterations in the higher levels of integration and disruptions in the integral functions and, therefore, of the conscience, where the former belong to the branch of classical neurology, and the latter to the branch of classical psychiatry [44].

Three different types of neuropsychiatric disorders can occur after brain injuries:Somatic (paralysis, balance deficit, etc.);Cognitive (aphasia, apraxia, amnesia);Emotional and behavioral (psychosis, depression).

In a patient with PD, the disintegration between the reptilian level of the motor control system consisting of the basal ganglia and the frontal cortex determines the typical forward curved gait and the deterioration of fine motility of the typical primate hand. The disintegration of the cortex of the anterior cingulate gyrus causes self-activating apathy. On the other hand, the breakdown of the areas of the orbit-ventromedial cortex causes deficits in social cognition and obsessive compulsive disorders. In contrast, the disintegration of the dorsal lateral cortex causes deficits in executive functions and, in an advanced stage, dementia. Despite this, in the case of the breakdown of the entire functional system, there is an autonomy of operation of each structural level, which explains why patients with PD, in case of danger, significantly improve their motor performance associated with a phasic dopamine secretion that allows signaling to carry out the task of protection against risk, evidencing how the basal ganglia assume a predominant role in the regulation of movement through the information that comes from the cortex [45].

## 4. Superior Functions Affected

### 4.1. Frontal Functions

The prefrontal cortex gives a degree of variety and complexity to the functions associated with this region and involves many processes, such as motor control, attention, working memory, and planning. The anterior part of the frontal lobes is closely linked to the limbic, motor, and sensory systems, and contributes to their regulation. The prefrontal cortex must be considered as part of an extensive attentional control network, particularly involving the parietal associative sensory regions subsequent and the mechanisms by which new behavior is automated. If the networks involved in this memory probably primarily cover those underpinning learning, a reorganization of representations is also suggested when behavior is automated. Motivational processes and reward processing thus acquire increasing importance in studies of the so-called high-level functions of the prefrontal cortex [46].

Dopaminergic neurons are closely associated with a vast forebrain territory, including the striatum and frontal cortex. They would intervene in regulating motor, motivational, and executive functions of the cortico–striatal–thalamo–cortical circuits; the fuzzy projection architecture likely cannot support the processing and storing of detailed information. On the other hand, it seems ideal for coordinating responses throughout the striatum, amygdala, hippocampus, and prefrontal cortex to salient stimuli, especially rewards. One of the main functions of the dopaminergic neurons of the ventral tegmental area is to develop motivation and generate behaviors adapted to the individual’s physiological state in the face of changes in the environment. By projecting almost exclusively to the limbic system, the prefrontal cortex, amygdala, hippocampus, and nucleus accumbens, these neurons modulate emotional activity, attention, and learning related to a new event [47,48].

A stressful or aversive experience can cause activations or inhibitions of dopaminergic neurons and changes in the level of dopamine released in projection regions. Therefore, dopaminergic neurons of the ventral tegmental area represent a heterogeneous population that can encode both reward and aversion [48].

The reduction in dopamine at the striatal level causes a dysfunction of the frontostriatal circuits, particularly the dorsolateral, in the early stages of the disease. A deficit in cognitive functions mainly related to the dorsolateral prefrontal cortex is the most common finding reported in studies investigating these aspects of PD [49].

### 4.2. Attention

Behavioral manifestations of dysexecutive disorders include attention deficit, which manifests itself in the inability of the patient to concentrate and maintain voluntary or automatic attention, and a tendency to be quickly and tirelessly attracted to irrelevant aspects of the environment [50]. Specifically, patients cannot voluntarily direct attention to interesting stimuli and events and have difficulty switching attention from one stimulus to another. Patients with PD generally show poor performance both in selective attention tests, in which the relevant information for the task must be selected from among various stimuli and inhibit the interfering ones, as well as in the Stroop Test [51], and in divided attention tests in which the attention resources are divided among several tasks to be performed simultaneously, as in the Trail Making Test [39,52,53].

### 4.3. Executive Functions

The term executive functions describes a set of psychological processes necessary to participate in adaptive and goal-oriented behaviors [54]. They include high-level processes such as planning, problem-solving, decision-making, cognitive flexibility, self-control, error detection, inhibition of automatic responses, and self-regulation [55,56]. All these processes allow the individual to coordinate the activities necessary to achieve a goal: formulate intentions, develop action plans, implement strategies for implementing these plans, monitor performance, and evaluate its results. Poor performance compared to control subjects is reported on executive tests [55,57].

The Wisconsin Card Classification Test is used to measure some executive functions, such as flexibility, task switching, or inhibition, as cognitive functions that are more affected in patients with cognitive impairment associated with PD [39,58,59]. A deficit in the ability to actively search for verbal material from one’s internal lexicon ultimately also produces poor performance in PD patients in tests of verbal fluency, both phonological and semantic [39,60,61,62]: the dysexecutive syndrome of the PD patient can result in poor production, with a lot of perseverance and violation of the rules.

The cognitive and emotional–behavioral disorders observed in PD can be possible manifestations of the disease or, secondarily, be induced by pharmacological therapies. This significantly affects the life and daily activities of patients and their families, often posing considerable difficulties to the specialist during the diagnosis and choice of treatment [63].

## 5. Psychiatric Disorders

### 5.1. Depressive Disorders

Mood alterations in PD may precede the onset of the disease and constitute a risk factor. Recent studies report an estimated prevalence of a depressive disorder in 40–50% of parkinsonian patients. The depressive symptoms in PD are different from those of primary depression: PD patients experience fewer feelings of guilt and self-reproach and more irritability, sadness, and concern about their health status. Mood fluctuations may be accompanied by “on-–off” motor fluctuations related to the duration of the effect of dopamine therapy, with a decrease in mood during the “off” state and an improvement during the “on” state [64]. As the disease progresses, many patients present the phenomenon of wasting, as in a reduction in the therapeutic efficacy of every single dose, which leads to the appearance of the fluctuation mentioned above phenomena between an “on” phase, during which the patient experiences a situation of well-being with complete or sufficient autonomy, and an “off” stage, in which there is no longer a response to the drug and the symptoms of the disease reappear. Depression in PD is not reactive to the condition of the disease but has a significant biological basis resulting from damage at the level of serotonergic, noradrenergic, and dopaminergic transmission [7]. The degree of depression is correlated with the severity of executive dysfunction, taken as an index of the level of functional deterioration of the frontostriatal circuits, and with the level of 3,4-dihydroxy-6-18 F-fluoro-L-phenylalanine uptake at the level of the striatum [65]. Episodic manic attacks in patients with PD are sporadic and generally related to side effects of therapy with dopamine agonists or high doses of levodopa [66].

### 5.2. Apathy

Apathy (a loss of interest and motivation to act) is present in approximately one third of PD patients and is independent of depressive disorders. Apathy in PD is self-activated apathy, characterized by the inability to self-activate thoughts or initiate actions due to altered circuits in the dorsolateral prefrontal cortex, the medial orbitofrontal cortex, and the gyrus cortex. Since one of the most critical functions of the basal ganglia is the self-activation of behavior guided by internal motivational impulses, apathy can be considered a pathology of the deactivation of the self-activation system that occurs after lesions of the basal ganglia. The activation of the cognitive and limbic areas of the frontal cortex in PD patients and the degree of apathy can predict the development of dementia: patients with apathy are subject to a higher rate of conversion to dementia than patients without apathy [67].

### 5.3. Alexithymia

Alexithymia (the difficulty of the subject to identify, describe, and communicate emotions and to distinguish between emotional experiences and physiological activation of emotions) is of particular interest for two reasons: in the general population, alexithymia is related to the presence of mood disorders, familiar in PD. Patients with PD present difficulties in processing emotional information, such as facial expressions and speech prosody, which are reflected in less emotional reactivity to emotional stimuli [68]. The alexithymia construct could be a clinical index capable of detecting difficulties in handling emotional information in patients with PD. Identifying the neuropsychological and psychiatric correlates of alexithymia is still preliminary, even in the general population, so the correlation between alexithymia and the clinical characteristics (motor and cognitive) of patients with PD could provide valuable information in the future.

### 5.4. Anxiety Disorders

Anxiety disorders are often associated with depressive disorders and may constitute a preclinical risk factor. The anxiety can present as panic attacks, phobias, generalized anxiety disorder, and somatic symptoms. As with depressive disorders, anxiety disorders can also be associated with “on–off” fluctuations related to the duration of the effect of the dopaminergic therapy, with a particular accentuation of symptoms during the “off” phases. A recent study reported a 43% prevalence of anxiety disorders (panic disorder: 9%; agoraphobia: 2%; social phobia: 7.9%; specific phobia: 17%; generalized anxiety disorder: 4%) in a sample of 127 patients with PD. Some of the dysfunctional neurotransmitter systems involved in anxiety in PD are mediated by serotonin, norepinephrine, dopamine, and GABA [69].

### 5.5. Obsessive Compulsive Disorders

The correlation between obsessive compulsive disorder and PD is due to the joint involvement of striatal structures, and the same neurotransmitters, dopamine and serotonin, are involved in obsessive compulsive and motor disorders [70]. An epidemiological study shows a higher frequency of this symptomatology than the control population, since obsessive compulsive symptoms seem to occur more frequently only in patients in advanced stages of the disease, especially patients with clinical motor onset on the left side. This phenomenon could suggest that the manifestation of obsessive compulsive symptoms is related to dysfunction of the frontostriatal circuits, especially in the right hemisphere, in line with what was found in patients with obsessive compulsive disorder. Some studies show that obsessive compulsive traits can manifest in the premorbid personality of PD patients: these are individuals with an inhibited nature, high moral rigor, and excessive self-control, and this personality type can induce a biochemical change in the cortex and basal ganglia over time and can be aggravated in a state of dopamine denervation by pulsatile supplementation of the same or by the addition of dopamine agonists, especially with an affinity for D3 receptors of dopamine at the mesolimbic level [71]. This personality type depends on an already existing alteration of the extrapyramidal system, so it is currently unclear whether the behavioral traits of the parkinsonian personality are related to dopaminergic dysfunction or frontal lobe, or are prodromal symptoms of the disease potentiate with the management of dopaminergic substitution in a denervated system [72].

### 5.6. Psychosis

The main psychotic symptoms reported in approximately 30% of PD patients with the advanced disease include hallucinations and delusions. Psychotic symptoms may be associated directly with the disease, drug therapy, or both. Hallucinations are predominantly visual and generally appear in the second half of the disease course. Hallucinations of insects or small animals predominate, while other times, they are fleeting visions of people, adults or children, and static and silent images; however, hallucinations are rare. Visual hallucinations, which are frequently observed in patients with an intact state of consciousness, materialize as visions of people, animals, or inanimate objects; they can become illusions, that is, distorted perceptions of existing stimuli, and have threatening content, inducing feelings of fear such as Charles Bonnet syndrome, in which patients report false perception of threatening forms of objects when there is a decrease in ambient brightness [73].

Delusions, represented by false beliefs that generally represent a misinterpretation of one’s experiences, are less frequent than hallucinations. Persecutory delusions are characterized by the patient’s mistaken belief that someone is threatening him. Sometimes delusions of jealousy may be observed, yet the less frequently reported misconceptions may be paranoia or jealousy. Misidentification delusions, such as Capgras syndrome, are very rare in PD patients, whereas they are more common when overt dementia occurs [74].

For a long time, it was believed that the psychotic phenomena in PD were effects of prolonged and high dopaminergic stimulation. Today, it is thought that the appearance of these phenomena is the effect of an interaction between some clinical characteristics and the therapy. For example, PD patients with visual hallucinations have more significant cognitive difficulties, particularly visuospatial, than patients without hallucinations.

### 5.7. Acute Confusional State

Confusional states (delirium) are more common in elderly PD patients. An alteration in the state of consciousness is associated with cognitive (memory deficit, spatiotemporal disorientation) and perceptual (hallucinations, illusions) disorders (Figure 5).

## 6. Neuropsychological Alterations

### 6.1. Amnestics and Non-Amnestics

Impaired cognitive function in non-demented PD patients consists of a broad spectrum of clinical deficits of varying severity that affect the amnestic and non-amnestic domains. The most-compromised cognitive functions are executive functions, information processing speed, visuospatial skills, language, and working memory. Administrative functions include the ability to plan, organize, initiate, and regulate behavior. It is mainly based on the frontostriatal circuit for prefrontal regions such as the dorsolateral prefrontal cortex and its connections with the basal ganglia. This frontostriatal circuit is a critical component not only in subcortical dementia in PD, but also in the mild cognitive deficit associated with PD [75].

Speech function is partially salvaged in non-demented PD patients. Nevertheless, a decrease in the information content of language, a reduced understanding of complex sentences, and a reduction in speech fluency can be observed. This deficit is mainly a consequence of the participation of the frontal lobe in forming concepts. Regarding memory, PD patients have difficulties learning new information, as evidenced by reduced free memory capacity, but can improve with semantic signals or recognition functions. Reduced registration and retrieval may be due to inattention or impaired executive function rather than coding deficits [76].

### 6.2. Memory Disorders

Memory disorders are often found in PD patients in the early stages of the disease [38,77]. In addition to marked alterations in working memory, PD patients show fewer deficits than Alzheimer’s disease patients in learning new information [62]. Until a few years ago, it was believed that patients with frontostriatal dysfunction, such as PD patients, performed better on recognition memory tests than free memory tests, suggesting that the storage process was intact and the retrieval process compromised [60]. Recently, this hypothesis of recovery deficit has been criticized by numerous studies that have demonstrated the existence of different memory profiles in PD patients [78]. PD patients’ results, assessed through multiple tests of episodic memory, performance on free retrieval tests, and performance on recognition or facilitated recall tests, are equally poor. A recent study reports deficits of the same level in PD patients in recognition, free memory, and prospective memory tests. The subgroup in a more advanced stage of the disease reports significant deficits in all previous trials [79].

Patients with PD do not have a “classic” memory profile but may have specific deficits in the individual processes that underlie the ability to memorize information; therefore, they may have different memory profiles and associations with different shapes concerning other cognitive functions. There is also the asymmetry of dopaminergic dysfunction (more significant on the side on which motor symptoms begin) about numerous cognitive functions that present asymmetric neural correlates. Concerning memory, for example, the left prefrontal cortex appears more involved in the storage phase while the right prefrontal is in recovery [80]. Until a few years ago, it was considered that the memory deficit in PD patients was almost exclusively due to the difficulty of recovery. This process has been mainly analyzed in this clinical population [81]. Many studies have focused on facilitated recovery in investigating the mechanisms that lead to recognizing a new or previously found element. This can be vivid (recollection), with details about the context in which it was found, or it can take the place of more intuitive (familiarity), without a real awareness of the context in which you were previously. Through different experimental procedures, the recognition process reports that the familiarity recognition modality is particularly deficient in patients with PD, while recall is much more comprehensive. This implies that the degree to which each patient with PD tends to depend more on one or the other modality will determine their degree of impairment in recognition tests.

Treatment with levodopa in patients in the early stages of PD facilitates cognitive flexibility (the ability to focus on multiple tasks alternately). Conversely, stopping levodopa treatment has a negative effect on cognitive flexibility but has a beneficial impact on reverse learning. This phenomenon is likely explained by the dopaminergic overdose hypothesis [82,83]. This hypothesis also suggests that the administration of levodopa replaces the decrease in dopamine in dysfunctional circuits (improving the cognitive functions connected to these circuits). Nevertheless, it causes an overdose of dopamine in mainly intact circuits (worsening the cognitive processes related to them).

The diminution of dopamine at the striatal level, initially in the dorsolateral frontostriatal circuit and later along the pathway, also in the orbital frontostriatal circuit, explains why the administration of levodopa is not directly correlated with improvement in cognitive performance. The clinical presentation of PD is generally asymmetric, indicating that the dopaminergic reduction is more significant in the ipsilateral hemisphere in expressing motor symptoms. This suggests that a certain dopaminergic overdose may occur not only in the frontostriatal orbital circuit but, more generally, in the cerebral hemisphere less affected by the dopaminergic reduction. There is a possible asymmetry of the dopaminergic decrease between the two hemispheres that can affect patients with PD at the level of the general cognitive profile [84], the attention and executive functions [85], and memory [86], while many cognitive functions have asymmetric neural correlates. However, few studies have investigated the direct relationship between dopamine agonist therapy and cognitive functions.

### 6.3. Working Memory

The working memory model describes a system with limited capacity that supports human thought processes while keeping information temporarily active, providing an interface between perception, long-term memory, and action. Working memory is also deficient in PD, both spatially [87] and verbally [39,88]. On examining these functions through tests of verbal and visuospatial amplitude [89], deficits are observed, especially in conditions that require manipulation of information compared to those that require simple maintenance of the data.

### 6.4. Decision-Making Processes

For years, neuropsychiatry and psychology have developed tasks to investigate executive functions connected to the ventromedial portion of the prefrontal cortex [54]. Traumatic or vascular damage in the ventromedial prefrontal site is associated with deficits in decision-making, described as “myopia or blindness for the future”; that is, the inability to evaluate and avoid the possible negative consequences of one’s actions [90]. Laboratory tests similar to gambling have been proposed to study the deficits in the decision-making capacity of patients with ventromedial prefrontal lesions. The Iowa Gambling Task (IGT) is the best known and used in the literature, and has been extensively used in PD patients.

### 6.5. Language

Instrumental functions such as language and praxis are rarely altered in PD, both in patients with dementia and non-demented patients [91]. Patients with PD have deficits in verbal fluency [61], but these can be interpreted as signs of executive dysfunction rather than a primary language deficit [92]. The generation of words effectively requires planning skills to be intact in semantic memory. Some PD patients may have naming deficits. These linguistic comprehension deficits appear to have multiple causes: a lack of cognitive flexibility and difficulty inhibiting response appear to compromise understanding of relative sentences.

In contrast, difficulty in verbal working memory appears to compromise the understanding of subordinates relative to ending long sentences. Oral work also seems to be related to more significant challenges for PD patients to understand metaphors. In 2007, Monetta and Pell showed a direct correlation between working memory performance and understanding of metaphors in PD patients [93]. The executive disorder can also be reflected in a linguistic deficit at the pragmatic level. There is a close correlation between measures of prefrontal functioning and pragmatic communication skills, measured through indices of the appropriateness of conversation, respect for turn, prosody, and proxemics [94].

### 6.6. Visuospatial Functions

Alterations in visuospatial functions are often reported in the literature, for example, in the Benton Orientation Judgment Test [39], but there is still much discussion about their genesis. Many tests are timed or involve motor dexterity factors, so even executive dysfunction can negatively affect performance on such tests (Scale Cube Drawing Test, WAIS-R) [95]. However, some recent studies [96] have confirmed a specific visual perception deficit in advanced PD, independent of executive deficit, due to neuronal degeneration at the retinal, subcortical, and cortical levels [97]. It should be noted that a marked visuospatial deficiency is considered a risk factor for the appearance of visual hallucinations [98].

### 6.7. Praxias

Apraxic disorders are poorly reported in PD [91]. Bilateral ideomotor apraxia is detected in 27% of PD patients compared to 75% of patients with progressive supranuclear palsy [99]. The degree of apraxic deficit, highlighted by the scores in the evaluation tests, is directly correlated with the degree of cognitive impairment and, in particular, executive dysfunction, confirming the crucial role of cortico-striatal circuits in the generation of apraxia. It should be noted that, in general, there is no correlation between scores on apraxia assessment tests and scores on scales that measure motor disability, such as the Unified PD Rating Scale [100]. This shows that ideomotor apraxia cannot be explained by the motor disability associated with PD and that these areas can be investigated independently [101]. A patient with apraxia presents a UPDRS-III with a higher score due to the inability to perform a motor activity, which must be considered.

On the other hand, a patient with PD and apraxia must already have dementia associated with Parkinson’s or have an overlapping Alzheimer’s disease, which can coexist in up to 30% of patients with the advancement of the disease and the age of the patients. However, the presence of apraxia of the extremities in PD remains controversial. Quencer et al. shows that PD patients are deficient compared to controls in a coin-operated finger rotation task. Even if they are able to perform a finger-tapping motor task as controls (thus not being bradykinetic, likely thanks to drug therapy), PD patients fail in the task of turning the coin [102]. This condition of apraxia of the extremities, therefore, does not appear to be attributed to the akinetic/rigid syndrome of PD.

### 6.8. Calculation

The executive deficit in PD also leads to cognitive impairments in numeracy. In a study comparing the performance of 15 non-demented PD patients with that of control subjects in several numerical skills tasks (quantity processing, arithmetic fact retrieval, written and mental math calculations, transcoding, computational span), the authors report that PD patients are deficient compared to controls in mental arithmetic and computational span tasks. However, these deficits are considered secondary to the main difficulties in working memory and sensitivity to interference in patients with PD [103].

## 7. Cognitive Disorder–Dementia

Among the non-motor disorders, cognitive disorder is mainly present in PD and includes a mild to moderate cognitive impairment (MCI) up to a clear picture of dementia. Dementia in PD occurs primarily in the advanced stages of the disease, with an estimated prevalence of approximately 80% twenty years after the onset of the disease [104].

Dementia associated with PD (PDA) as a Parkinson’s–dementia complex (PD-D) represents 3–4% of the forms of dementia present in the general population [105]. The incidence of PDA is 2.5 overall and also increases considerably with age. Regarding the incidence of dementia with Lewy bodies (DLB) and PDA combined, it ss 5.9. The patients with MCI were younger at the onset of symptoms than those with PDA, presenting more hallucinations and cognitive fluctuations. Men have a higher incidence of MCI than women across the age spectrum [106]. Several studies agree that advanced age and the severity of motor symptoms are risk factors for neuropsychiatric symptoms in patients with PD; the additional risk factors identified are the akinetic-rigid onset, the presence of hallucinations, and psychotic disorders. Movement disorders working groups defined the diagnostic criteria for PD in 2015 [5] and, in 2017 [107], the clinical diagnostic criteria for possible and probable PD-D, and proposed a practical approach for diagnosis. The fundamental criteria include the diagnosis of PD, the presence of a syndrome of major cognitive disorder (DSM-V) that develops insidiously and progressively in the context of overt PD, and the alteration of more than one cognitive domain with alteration of the activities of daily living, which implies a reduction in autonomy. In the first years of PD with motor impairment, it is possible to identify an MCI responsible for negatively impacting the quality of life and the caregiver’s stress. Mild cognitive impairment is a potential developmental stage between normal cognitive status and dementia. Diagnostic criteria have been proposed by Petersen, and include the anamnestic response of a cognitive disorder, the presence of a deficit in at least one cognitive test, and preserved autonomy in the performance of activities of daily living [108]. There is an increase in the diagnoses of MCI, and data from the literature show that about 10–15% of patients with MCI per year progress to dementia [109] (Figure 6).

## 8. Neuropsychological–Psychiatric Evaluation

Numerous studies have demonstrated the importance of neuropsychiatric–psychological (NP) evaluation in diagnosing extrapyramidal syndromes for a correct nosographic and prognostic classification because cognitive disorders, previously considered auxiliary in the clinical picture, have a considerable weight in the reduction in the quality of life of patients [110].

An in-depth examination of cognitive functions makes it possible to make a differential diagnosis between isolated cognitive alterations, frequently present in parkinsonian patients, and actual dementia associated with PD or added as an independent entity.

The qualitative analysis of the NP impairment profile will, in turn, differentiate forms of dementia not associated with Parkinson’s from those related to clinical parkinsonian conditions, as well as real dementias from pseudo-depressive dementias, and guide the clinician in the correct therapeutic management of the patient. Qualitative analysis of the NP deficit in representative samples of patients with extrapyramidal diseases has contributed to the knowledge of the mechanisms regulating the frontal–striatum circuit, clarifying the relationship between motor disorders and cognitive–behavioral disorders and bringing to light the differences. Therefore, recent studies have attempted to interpret the cognitive condition in PD in anatomical and psychopharmacological terms and neuropsychiatric terms.

A neuropsychological screening battery must be able to collect information on various cognitive areas. PD screening must include selected tests based on their proven sensitivity to demonstrate deficits in mainly impaired cognitive functions in these patients. The Mini-Mental State Examination (MMSE) provides guidance on the possible presence of generalized cognitive impairment but does not allow a formal assessment of individual cognitive abilities, and is not as sensitive for evaluating executive functions at the onset of impairment. The Montreal Cognitive Assessment (MoCA) is more sensitive to assessing executive functions. The MMSE is a screening test for global cognitive function (temporal, spatial orientation, memory, attention, calculation skills, language, and practical skills).

On the other hand, the Milan General Dementia Assessment (MODA) and the Mild Deterioration Battery (MDB) allow us to compare the cognitive performance of groups of patients suffering from various neurological diseases. The MODA studies multiple cognitive areas, such as attention, intelligence, memory, language, and visuospatial skills. The MDB provides information on the functional efficiency of various cognitive domains such as memory skills (episodic short- and long-term memory for verbal material, short-term memory for visual perceptual material, semantic memory), constructive praxis skills, language and language skills, and conceptual and logical reasoning. The MDB is not only capable of discriminating patients with dementia from normal subjects, but is capable of identifying differential profiles of neurological deterioration in groups of patients affected by dementia syndromes of various etiologies: Alzheimer’s disease, vascular dementia, and PD.

The dysexecutive syndrome in PD is constituted by the inability of these patients to develop adaptive cognitive–behavioral patterns in response to new and challenging environmental conditions and to guide specific behavior. The most commonly used screening tests are:Frontal Assessment Battery (FAB), Wisconsin Card Classification Test (WCST), Figure Classification and Recreation Test, Tower of London Test, Dysexecutive Syndrome Behavioral Assessment (BADS).For assessing long-term declarative memory for verbal material, the tests consist of remembering and recognizing passages in prose and word lists: Babcock’s Tale, wordlist learning, and Rey’s 15 Word Test.Long-term visuospatial memory is assessed using the Corsi Test. The Corsi Test is also used for long-term declarative memory.Short-term memory is evaluated through the Digit Span Test.We can use the attentional matrices and the Trail Making Test regarding attention.Frontal functions are assessed through the Phonemic and Semantic Fluency Tests and the FAB.

The motor function of the basal ganglia has much interaction with motivational and affective tasks that interact with the limbic and prefrontal systems to perform specific tasks. Its state of rest is that of “doing nothing” and changes according to cortical afferents concerning environmental stimuli to release certain types of influence that manage to release movement in a modulated way and to perform motor tasks previously learned or developed through anagram-specific motors that are recorded in the basal ganglia, as well as the decision to perform only one or several motor tasks at the same time, and the possibility of stopping all the functions that are being carried out to start a new task or continue doing the preselected task, which is considered as the inertia of the compromise. Under different environmental circumstances, the motivational need to execute these motor anagrams has a close affective and volitional relationship with the limbic system and the prefrontal cortex. In this sense, the basal ganglia and their dysfunction are related through specific circuits with mechanisms of behavior, memory, and decision-making that are related to the cognitive and affective processes associated with this type of movement anagrams; therefore, its involvement has a direct and indirect impact on these types of neuropsychiatric circumstances.

## 9. Treatment

Treatment continues to be symptomatic but should not consist of only a dopamine substitution, as multiple therapeutic options are required: pharmacological, physiotherapeutic, surgical, neuropsychological, and psychiatric. Some of the non-motor symptoms, such as depression or rapid eye movement (REM) sleep behavior disorder, smell disturbances, or intestinal constipation, often occur in the prodromal phase before motor symptoms (premotor symptoms) [111].

The syndrome of dopaminergic dysregulation in PD is a particular behavioral disorder. Attraction and addiction characterize it as antiparkinsonian treatment, particularly dopa, which is used beyond a justified need. In PD, the mesocorticolimbic dopaminergic system is partially preserved; dopaminergic treatment, which aims to stimulate the flawed nigrostriatal dopaminergic system, triggers hyperactivity of the mesocorticolimbic system. This hyperstimulation of the mesocorticolimbic dopaminergic pathway is responsible for various neurobiological disorders with sensitization of the ventral striatum and dysregulation of controls of prefrontal origin. The sensitization of the ventral striatal dopaminergic system will determine changes in neuronal plasticity that will lead to biological and cellular changes responsible for an unusual attraction towards dopa. Dopaminergic treatment becomes both the cause of this process and the consequence by triggering the compulsive need for this substance through auto-training. This will be used more to avoid the clinical manifestations of the deficiency than for the benefit that could be obtained. Far from cognitive control, addiction to dopaminergic drugs becomes a reflex phenomenon of a subcortical nature, which the parkinsonian patient has difficulty controlling. This syndrome will be responsible for mood disorders, impulse control disorders, and repetitive and compulsive acts that result in the patient’s social isolation. Treatment is complex, and brain stimulation of the subthalamic nucleus can sometimes represent a means of therapeutic management by reducing the treatment it provokes [112,113].

The approach to treating psychiatric symptoms in PD involves pharmacological and non-pharmacological therapies. Good clinical efficacy has been reported in several controlled clinical trials for the tricyclic antidepressants (TCAs) desipramine and nortriptyline, the selective serotonin and norepinephrine reuptake inhibitor (SNRI) venlafaxine, and the selective serotonin reuptake inhibitors (SSRIs) citalopram, sertraline, and paroxetine. Among non-pharmacological strategies, cognitive behavioral therapy (CBT) appears to be the most promising approach; the use of transcranial magnetic stimulation (TMS) has also produced encouraging results, while DBS appears to be associated with worsening mood, apathy, and increased risk of suicide [114,115,116].

## 10. Conclusions and Perspectives

PD is a multisystemic condition that involves not only the nervous system but also the skin, the gastrointestinal system, and the autonomic system, among others, and in the nervous system, not only motor function is affected; on the contrary, there are non-motor symptoms that can start years before. There is evidence of non-motor symptoms that can appear during the disease, and that can sometimes be more disabling and annoying than the motor symptoms, in addition to the fact that they can aggravate the latter. Of these non-motor symptoms, neuropsychiatric problems are the ones that have a fundamental role in the disability and severity of the disease.

Among the neuropsychiatric symptoms that appear even before the appearance of motor symptoms are depression that manifests itself more than as a melancholic depression, as an apathetic depression, more difficult for the patient and family to perceive, and that consists of environmental demotivation and reduction in recreational and social activities and that can occur in up to 40% of patients during the disease. On the other hand, cognitive disorders appear, sometimes from the onset of the disease, in between 20 and 35% of patients, depending on the series, and become evident in up to 40% of patients, mainly affecting care and working memory, and associated with a dysexecutive syndrome, which can generate a reduction in the patient’s quality of life and can progress to the range of major neurocognitive disorder with social dependence in up to 80% of patients between 15 and 20 years of evolution of the disease, generating severe disability, a significant reduction in the quality of life, and exhaustion of caregivers and family members. Associated with cognitive deterioration, psychiatric problems such as psychosis may also occur in 10 to 30% of patients with advanced PD due to the disease or its association with medications indicated for managing the condition. Anxiety occurs in 30% of patients during the course, and can sometimes be a fluctuating non-motor symptom associated with taking drugs and become severe.

It is imperative to detect and treat this type of problem because psychosis is one of the main predictors of the need for institutionalization of patients and is associated with a reduction in the life expectancy of patients to no more than one year. Anxiety and depression indirectly aggravate parkinsonian motor symptoms. Only increased depression and cognitive impairment have been associated with a severe disability on the ADL scale of the UPDRS-II in 37% of cases. However, sometimes patients do not self-perceive them, and neither do family members. If the doctor does not ask them directly, they can be ignored and continue contributing to the worsening of the condition in patients [117,118].

Taking major neurocognitive disorders into account, there is a two- to six-fold increased risk of developing this condition in PD than in people of the same age without PD [119]. In addition, in the rigid-akinetic variants of PD, or those with postural instability and gait problems from the beginning, cognitive impairment usually appears earlier in the evolution and sometimes during the first years of the onset of the symptoms. Motor symptoms force a differential diagnosis of dementia with Lewy bodies, closely associated with a complex called dementia and PD.

As seen from these observations, it is imperative to form transdisciplinary teams to detect, treat and rehabilitate patients with the Parkinsonian condition’s neurodegenerative process, including a group of neuropsychologists who can detect these problems with special instruments and can apply these tests in an evolutionary way to visualize the speed of deterioration or the stability of the patients. It is also imperative to be able to establish cognitive re-habilitation programs supported by psychiatrists or neuropsychiatrists to manage affective, anxiety, and psychotic symptoms, and neurologists who, in addition to managing motor aspects and other non-motor symptoms, also help to pharmacologically manage the cognitive deterioration of the patients when necessary, without forgetting the geriatricians and internists who help as integrating axes of care to manage all the comorbidities that can aggravate and accelerate the neurodegenerative processes in patients such as diabetes mellitus, metabolic syndrome, and arterial hypertension, among many others.

## Figures and Tables

**Figure 1 brainsci-13-00508-f001:**
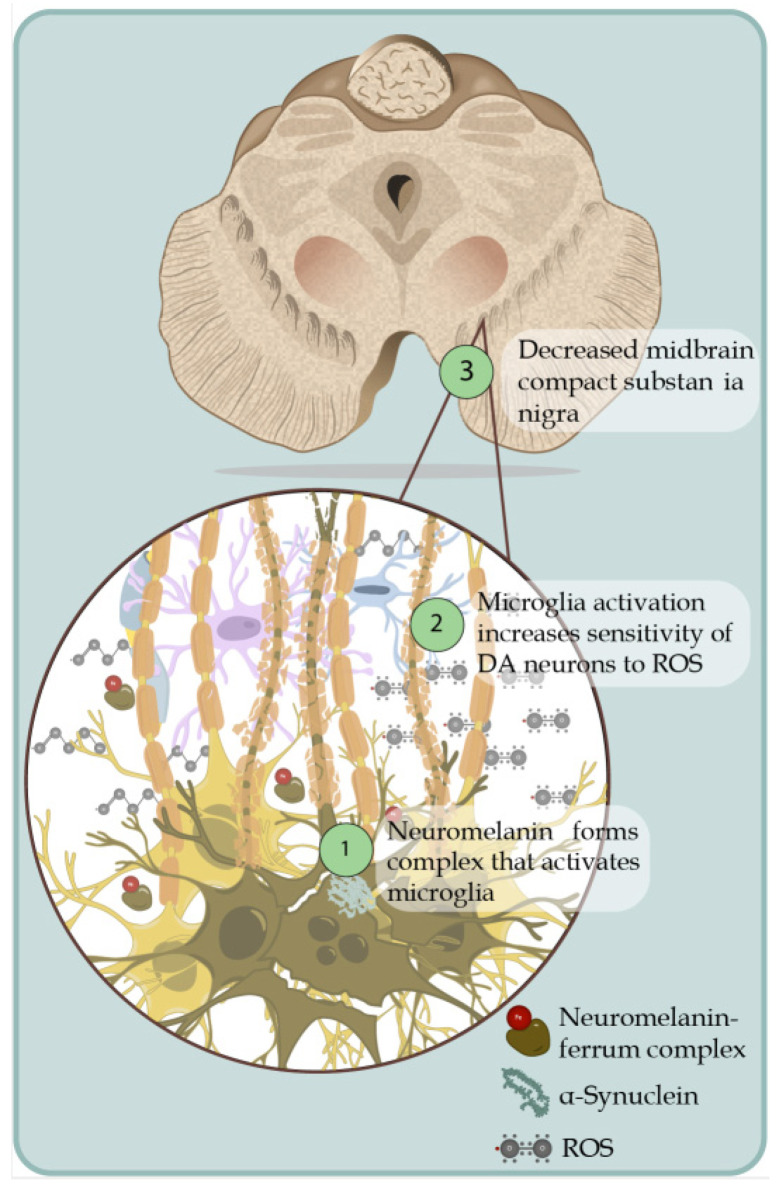
Death of dopaminergic neurons. Schematic representation of dopaminergic neuronal death associated with the generation of ROS in an uncontrolled way. 1. The pigment that gives color to dopaminergic cells is due to the oxidation of cysteinyl-DA products (dopamine that is not adequately removed works as ROS-generating molecules), products that interact with Fe+ forming a complex that activates the microglia and thereby generates the production of pro-inflammatory cytokines and the production of ROS, which induce the formation of alpha-synuclein fibrils. 2. Activation of microglia by the release of neuromelanin from pars compacta causes an increase in the sensitivity of dopaminergic neurons to oxidative stress; the dopaminergic neurons of the compact substantia nigra are more susceptible to ROS, which is why they more easily cause the death of this neuronal subpopulation in areas such as nuclei A9. 3. Decreased midbrain compact substantia nigra.

**Figure 2 brainsci-13-00508-f002:**
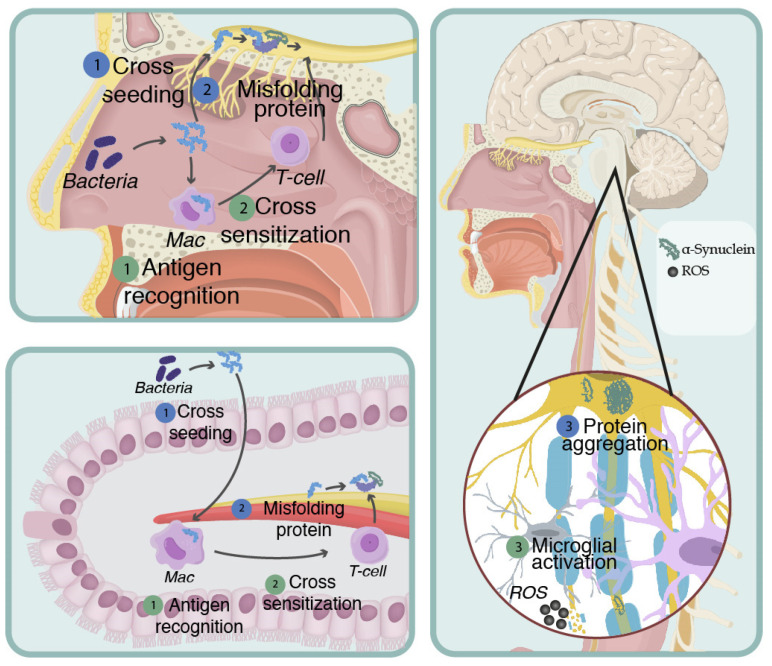
Mapranosis. Schematic representation of Friedland and Trudler’s proposal to explain neurodegeneration. Amyloid is secreted by bacteria resident in the nasal and intestinal epithelium, which causes neuroinflammation in two ways: 1. Misfolded Protein 1.1. Bacterial amyloid can cross seed with host beta-amyloid and alpha-synuclein. 1.2. This cross seeding induces the wrong folding of the alpha-synuclein due to prionic behavior; therefore, it extends through the vagus and olfactory nerve. 1.3. The new poorly folded endogenous proteins it disseminates reaches the CNS, causing neuronal death. 2. Induction of oxidative stress and neuroinflammation. 2.1. Bacterial amyloid is capable of inducing inflammation and oxidative stress. 2.2. Subsequently, there is cross sensitization against endogenous amyloid, which causes the immune system’s abnormal recognition of its epitopes, mounting a response directed against beta-amyloid and alpha-synuclein in the host’s CNS. 2.3. The sensitized immune cells reach the CNS through the hematogenous pathway, crossing the BBB, and activating the microglia, causing neuronal death.

**Figure 3 brainsci-13-00508-f003:**
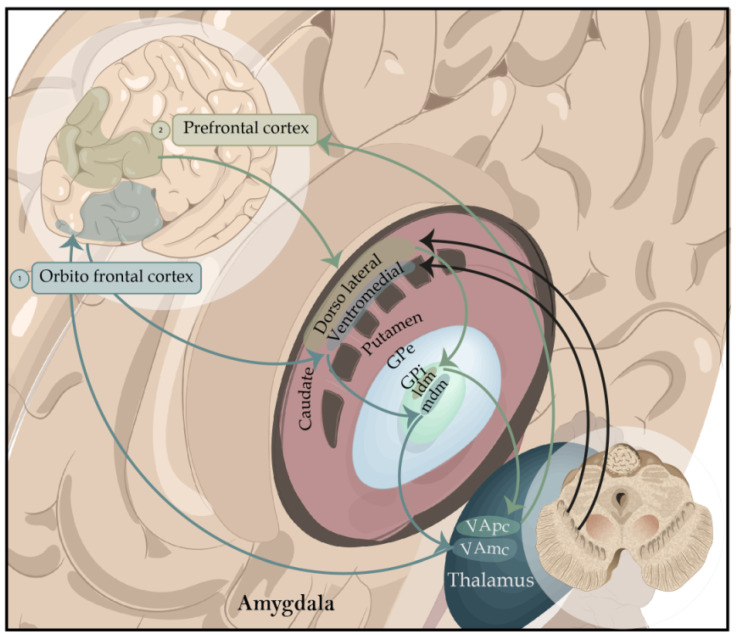
Representation of the basal ganglia focused on cognitive functions. Ventrolateral circuit. 1. It begins in the ventrolateral frontal cortex, which is the first node of interaction with the posterior regions. It’s in charge of low-level control processes and involved in stimulus-driven recall and information retrieval from long-term memory. Lateral mediodorsal circuit. 2. It begins in the mediolateral cortex of the frontal cortex. It’s in charge of high-level executive processes and involved in the manipulation and monitoring of care and is in charge of creating organizational strategies. Abbreviations: GPe, globus pallidus pars externa; Gpi-ldm, lateral dorsomedial globus pallidus pars interna; GPi-mdm, medial dorsomedial globus pallidus pars interna; VApc, ventroanterior thalamic nucleus parvocellular part; VAmc, medialventroanterior thalamic nucleus magnocellular part.

**Figure 4 brainsci-13-00508-f004:**
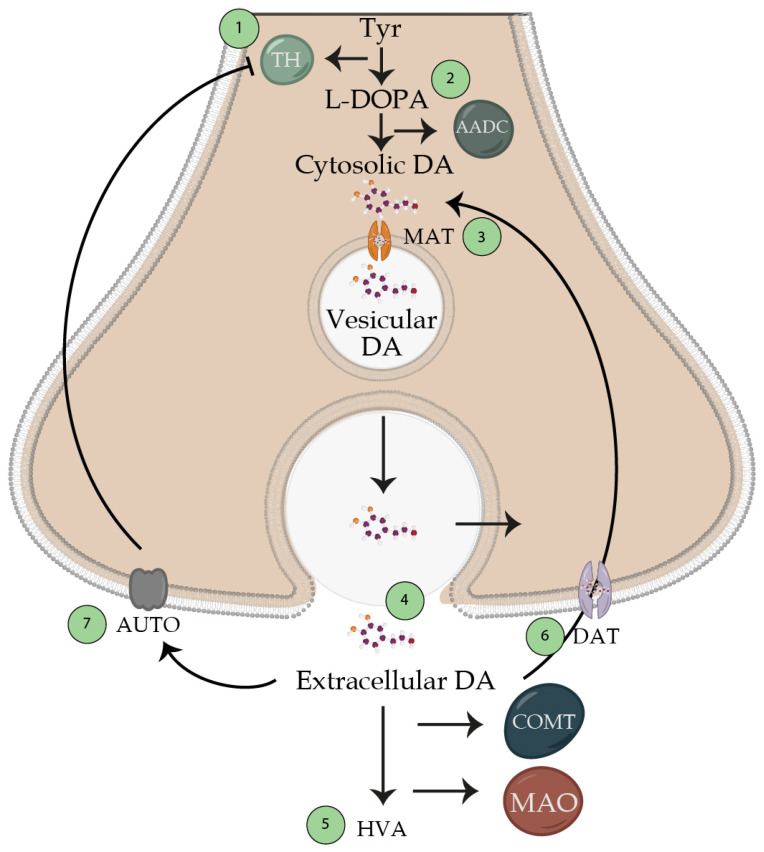
Dopamine synthesis. Schematic representation of the synthesis of the neurotransmitter dopamine. 1. Produced from the amino acid tyrosine, which passes through tyrosine hydroxylase, which generates L-DOPA. This enzyme is the one that limits the production of dopamine (which is inhibited by high concentrations of its substrate tyrosine). 2. Transformation of L-DOPA into dopamine by the aromatic amino acid decarboxylase enzyme, which is found in a cytosolic way and is not active for the release with the action potentials. 3. Transport of cytosolic dopamine to vesicles via the vesicular monoamine transporter enzyme. 4. The potential action releases dopamine to the synaptic cleft, so it reaches its receptor and generates its function. These nerve terminals will be found in the circuits already mentioned in the text; however, we can find a tonic release at a frequency of 5 Hz, and another 10–15 Hz interburst release. 5. Extracellular dopamine will be metabolized by catecholamine O-methyl transferase and mono-amino oxidase towards homovalinic acid. 6. Presynaptic receptors will recapture unmetabolized dopamine for reuse. 7. Unmetabolized dopamine may bind to presynaptic receptors to block tyrosine hydroxylase. Abbreviations: Tyr, tyrosine; TH, tyrosine hydroxylase; AADC, aromatic acid decarboxylase; MAT, monoamine transporter; DAT, dopamine transporter; MAO, monoamine oxidase; COMT, catecholamine O-methyl transferase.

**Figure 5 brainsci-13-00508-f005:**
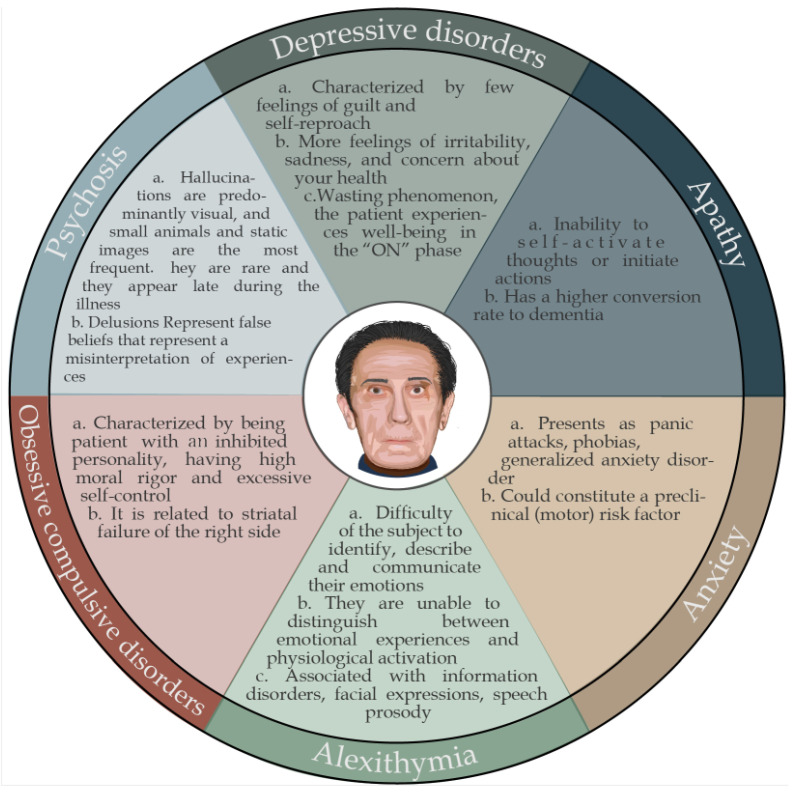
Summary of characteristic neuropsychiatric alterations in Parkinson’s disease.

**Figure 6 brainsci-13-00508-f006:**
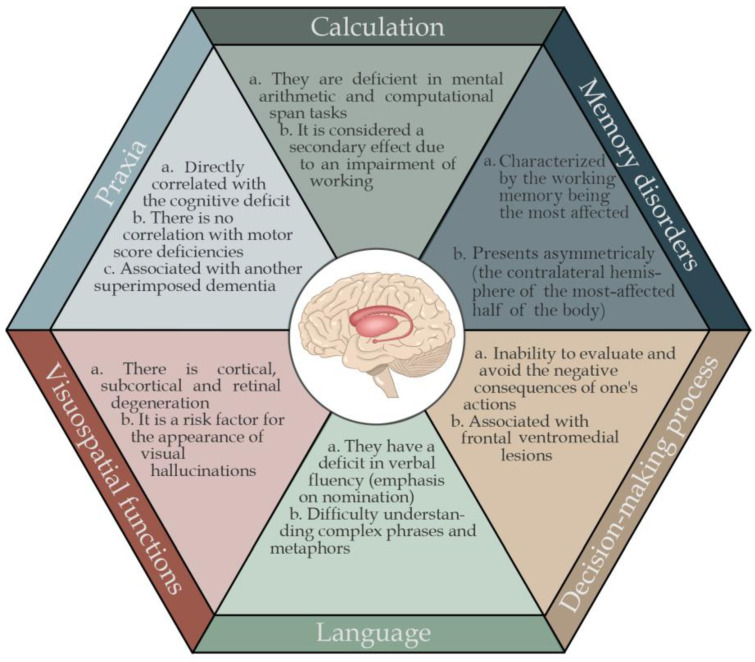
Summary of the characteristic cognitive alterations in Parkinson’s disease.

## Data Availability

Not applicable.

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
