# Peer review of "Neurocognitive Psychiatric and Neuropsychological Alterations in Parkinson’s Disease: A Basic and Clinical Approach"

_brainsci, 2023, doi:10.3390/brainsci13030508_

Round 1

Reviewer 1 Report

Authors have done an outstanding job of reviewing the literature about the pathophysiology of Parkinson’s disease and its clinical neurocognitive and psychiatric manifestations

Author Response

Thank you for reviewing our work and for your comments.

We check all the text for grammar and spelling.

Reviewer 2 Report

The paper is interesting and largely well-written. It is evident the effort of the authors to include much information. Likewise, figures are illustrative and provide additional information to understand the paper. I have a few questions and comments.

Please carefully revise the acronyms used throughout the manuscript; some words are used after they were already defined (e.g., Parkinson's disease, reactive oxygen species)

The abstract appears a little informative. The objective seems unclear; the authors could emphasize their paper's contribution to the knowledge of non-motor symptoms of Parkinson's disease.

The authors should write their manuscript's objective in the introduction section's final paragraph.

The introduction section could be more focused on non-motor symptoms, especially in neuropsychological and neuropsychiatric manifestations.

Section 2 (Genetics) is unnecessarily large; it could be shortened or (even) deleted because it seems irrelevant to the manuscript topic. Actually, only the last sentence of this section appears to be related to the manuscript theme (The incidence of this disorder in western countries is around 200 cases per 100,000 people (depending on the country), most of them over 60 years of age. The main symptoms of PD involve movement and include tremors, stiffness, bradykinesia, and postural disturbances. Parkinson's Disease is not just a motor disorder since many patients show deficits in cognitive function up to dementia with severe impairment of memory, abstract thinking, and language. The presence of affective and behavioral disorders varies from 12 to 90% of cases and frames PD as a neuropsychiatric disorder).

Figures 5 and 6 are illustrative. However, the authors should add a conclusion section to their manuscript to summarize their work (it could be a brief paragraph).

 Please see attached PDF for minor issues (typos, grammar mistakes, etc.)

Author Response

Thank you for reviewing our manuscript and for all your comments.

Acronyms throughout the text were revised and corrected.

Added a graphical abstract to strengthen the summary.

A paragraph focused on the non-motor symptoms of PD was added to the introduction section.

Added the objective of the manuscript at the end of the introduction section.

A part of the genetics section was eliminated, which was left with the name of pathophysiological factors.

Added a section on conclusions and perspectives.

We check all the text for grammar and spelling.

We made all the recommendations made in the attached pdf file.

Reviewer 3 Report

- Subsection on "recommendations" for treatment

- Subsection on "limitations"

- Subsection on "conclusions"

Author Response

Thank you for reviewing our manuscript and for your comments.

Added treatment section.

Added conclusions and perspectives section.

We check all the text for grammar and spelling.

Reviewer 4 Report

The authors present a comprehensive history and overview of the pathophysiology and histopathology of Parkinson disease, along with an expansive description of the different functional outcomes of neuronal circuitry impairments observed in patients with PD. I anticipate that this review article will serve as an important reference for clinical research on the topic of early diagnosis, treatment outcome measurements, and subtyping of PD-related conditions.

The detailed explanation of the interaction between epithelial-associated bacteria, amyloid secretion, and dopaminergic cell loss is described well. The figures are professionally prepared and helpful for understanding the text.

Line 17: I recommend changing "pathological hallmarks" to "histopathological hallmarks". I would also recommend describing the loss to the dopamine neurons of the substantia nigra pars compact as the first thing, because it is the loss of dopamine which is the pathology of PD. The loss of neuromelanin is a consequence of decreased dopamine synthesis and is not considered to be causative in PD.

Line 36: In American English the historic term used by James Parkinson is usually referred to as "shaking palsy", not "agitation paralysis". I believe Parkinson used the Latin "paralysis agitans". Authors can change at their discretion and this is not a necessary change.

Please expand on subsection 4.1 to describe the exact functions that are impaired by loss of dopamine to the orbitofrontal region of the brain, as you have done with the other subsections of section 4. Are there neurological or psychiatric tests that can be used to assess such a deficit?
https://academic.oup.com/book/8333/chapter-abstract/153992884?redirectedFrom=fulltext
https://pubmed.ncbi.nlm.nih.gov/20728457/

Author Response

Thank you for reviewing our manuscript and for all your comments.

Change "pathological hallmarks" to "histopathological hallmarks".

Describe first the loss of dopamine in the abstract section.

Change the term for “ paralysis agitans”.

Add more information in subsection 4.1 describing frontal functions.

We check all the text for grammar and spelling.

Round 2

Reviewer 2 Report

The authors performed all recommendations, and the manuscript is notably improved. My only observation is using the acronym for Parkinson´s disease (PD) in the new sections (sections 9 and 10).